# Dietary Supplementation of Foxtail Millet Ameliorates Colitis-Associated Colorectal Cancer in Mice via Activation of Gut Receptors and Suppression of the STAT3 Pathway

**DOI:** 10.3390/nu12082367

**Published:** 2020-08-07

**Authors:** Bowei Zhang, Yingchuan Xu, Shuang Liu, Huan Lv, Yaozhong Hu, Yaya Wang, Zhi Li, Jin Wang, Xuemeng Ji, Hui Ma, Xiaowen Wang, Shuo Wang

**Affiliations:** 1Tianjin Key Laboratory of Food Science and Health, School of Medicine, Nankai University, Tianjin 300071, China; bwzhang@nankai.edu.cn (B.Z.); Xuyingchuan_6@163.com (Y.X.); 2120171314@mail.nankai.edu.cn (S.L.); lvhuan@nankai.edu.cn (H.L.); yzhu@nankai.edu.cn (Y.H.); 18829349424@163.com (Y.W.); lizhi3204@126.com (Z.L.); wangjin@nankai.edu.cn (J.W.); jixuemeng@nankai.edu.cn (X.J.); mahui2018@mail.nankai.edu.cn (H.M.); 2College of Food Science and Engineering, Shanxi Agricultural University, Taigu 030801, China; wwxw11@163.com; 3Shanxi Functional Food Research Institute, Taigu 030801, China

**Keywords:** foxtail millet, grain, colorectal cancer, colitis, aryl hydrocarbon receptor (AHR), G-protein-coupled receptor (GPCR)

## Abstract

Coarse cereal intake has been reported to be associated with reduced risk of colorectal cancer. However, evidence from intervention studies is absent and the molecular basis of this phenomenon remains largely unexplored. This study sought to investigate the effects of foxtail millet and rice, two common staple grains in Asia, on the progression of colitis-associated colorectal cancer (CAC) and define the mechanism involved. In total, 40 BALB/c mice were randomized into four groups. The Normal and azoxymethane/dextran sodium sulfate (AOM/DSS) groups were supplied with an AIN-93G diet, while the millet- and rice-treated groups were supplied with a modified AIN-93G diet. Compared to the AOM/DSS-induced CAC mice supplemented with rice, an increased survival rate, suppressed tumor burden, and reduced disease activity index were observed in the millet-treated group. The levels of IL-6 and IL-17 were decreased in the millet-treated group compared to both the AOM/DSS and AOM/DSS + rice groups. Millet treatment inhibited the phosphorylation of STAT3 and the related signaling proteins involved in cell proliferation, survival and angiogenesis. These beneficial effects were mediated by the activation of gut receptors AHR and GPCRs via the microbial metabolites (indole derivates and short-chain fatty acids) of foxtail millet. Moreover, millet-treatment increased the abundance of *Bifidobacterium* and *Bacteroidales_S24-7* compared to the rice-treated mice. This study could help researchers to develop better dietary patterns that work against inflammatory bowel disease (IBD) and for CAC patients.

## 1. Introduction

According to the latest data from the American Cancer Society, colorectal cancer (CRC) is the third most diagnosed and second most deadly cancer, seriously threatening human health [1]. In recent years, the morbidity and mortality of CRC has increased continuously, especially in some developing countries [2]. Inflammatory bowel disease (IBD), a type of inflammatory condition of the gastrointestinal tract that is mediated by abnormal immunity and has a tendency to relapse, has been reported to be an important inducer of CRC [3,4]. Epidemiological studies have suggested that IBD patients exhibit a higher risk of CRC and that the incidence of cancer is positively correlated with the duration of IBD [4,5]. The increased pro-inflammatory cytokine release and altered signaling pathways associated with IBD play vital roles in the development of colitis-associated and sporadic CRC [2]. Because prevention is better than a cure, intervention at the points of intestinal inflammation and early-stage tumorigenesis is of great significance in controlling the morbidity and mortality of CRC.

G-protein-coupled receptors (GPCRs), including GPCR41 and GPCR43, are crucial gut receptors that regulate intestinal immune functions [6]. The aryl hydrocarbon receptor (AHR), another gut receptor, also contributes considerably to regulating the immune response at barrier sites [7]. Inactive AHR is kept in the cytoplasm and can be directly activated by dietary molecules; this AHR then translocates into the nucleus and regulates target gene expression [8]. GPCR 41, GPCR43, and AHR play important roles in maintaining the integrity of the intestinal epithelial barrier, regulating the immune response, suppressing the production of pro-inflammatory cytokines and the expressions of GPCR43 and AHR decrease in IBD patients compared to healthy people [9,10]. On the other hand, the signal transducer and activator of transcription 3 (STAT3), which dimerizes and translocates into the nucleus after activation, functions as a crucial link between inflammation and cancer [11]. STAT3 can be activated by a number of cytokines, such as IL-6, IL-17, and tumor necrosis factor-α (TNF-α). Moreover, the continuous activation of STAT3 in tumor cells can enhance the expression of genes associated with cell survival, proliferation, and growth, including B-cell lymphoma-2 (Bcl-2), proliferating cell nuclear antigen (PCNA), and vascular endothelial growth factor (VEGF) [11,12]. Thus, the inhibition of STAT3 phosphorylation is a critical target for preventing the development of CRC. In animal studies, the combination of mutagenic agent azoxymethane (AOM) and dextran sodium sulfate (DSS) has widely been used in mimicking the development of colon cancer [13]. To establish the colitis-associated colorectal cancer (CAC) model in mice, AOM was first administrated to induce carcinogenesis, and then the mice were continuously exposed to the inflammatory stimulus of DSS to increase the incidence of colon tumorigenesis.

In recent years, researchers have highlighted the relationship between diet, gut microbiota, and CRC. Multiple studies have indicated that disorder in the gut microbiome structure can induce CRC, and the composition of the microbial community can be significantly impacted by one’s diet [14]. Moreover, the gut microbiota can ferment dietary residues that are resistant to digestion by enteric enzymes. These microbial metabolites, such as tryptophan metabolites and short-chain fatty acids (SCFAs) can regulate the host’s health through the activation of gut receptors [2,6]. Cereal foods are typically indispensable sources of energy and dietary fiber intake, and constitute the backbone of human diets all over the world. Epidemiological studies have suggested that a high whole grain or coarse cereal intake contributes to a reduced risk of CRC, which could be attributed to a high content of dietary fiber [15,16]. Researchers used chemically induced animal models to show that dietary cereal intake, such as the intake of barley and wholegrain oats, reduces the risk of colorectal cancer [17,18,19]. However, there is still a lack of strong evidence from intervention studies, on whether other types of cereals can reduce the risk of CRC. The intestinal mechanism also remains unclear.

Foxtail millet (*Setaria italica*) is widely cultivated in Asia and Africa, and is the most consumed coarse cereal in China [20]. Rice is the most common staple grain, it is widely consumed in Asia, and is usually processed into refined grain. The nutritional qualities of millet are superior to those of rice, as the contents of protein, fat, and dietary fiber in foxtail millet are richer than those in rice, and millet also contains less starch than rice [21]. In addition, millets are rich in tryptophan, which is an essential human amino acid primarily acquired through dietary intake [22].

Rice and millet are both primary sources of staple foods in China and the rest of Asia. Differences in the composition of nutrients in millet and rice can shape the structure of the intestinal flora and the composition of metabolites after human intake. Whether long-term dietary intervention can have effects on the development of CAC remains to be determined. Therefore, we compared the effects of millet and rice on CAC and explored the underlying molecular mechanisms involved.

## 2. Materials and Methods

### 2.1. Chemicals and Materials

Azoxymethane (AOM) was obtained from Sigma-Aldrich Corp. (St. Louis, MO, USA), and dextran sulfate sodium (DSS, MW 36–50 kDa) was obtained from MP Biomedicals LLC (Santa Ana, CA, USA). TRIzol^TM^ Reagent (Thermo Fisher Scientific), LunaScript^TM^ SuperMix Kit (New England BioLabs), SYBR qPCR Master Mix (ChamQ^TM^ Universal) were used for RT-qPCR analysis.

### 2.2. Animals

Forty specific-pathogen-free (SPF) male BALB/c mice (4 weeks old) were supplied by SPF Biotechnology Co., Ltd. (Beijing, China) and raised under standard laboratory conditions of 25 ± 2 °C and 50 ± 5% relative humidity, with 12 h light–dark cycles. The mice were housed in groups of five per cage. They were kept in a room isolated from all other ongoing animal experiments and handled only by the primary investigators. The mice were supplied with a customized rodent diet based on AIN-93G and had free access to water. All protocols for the animal experiment were approved by the Institutional Animal Care and Use Committee of Nankai University and carried out in compliance with the national ethical guidelines for laboratory animals.

### 2.3. Induction of Colorectal Cancer and Experimental Design

Forty mice were randomly allocated into four groups (NM, AOM/DSS, AOM/DSS + millet, and AOM/DSS + rice, *n* = 10). The colitis-associated CRC model was induced by AOM/DSS [23] and the experimental protocol was shown in Figure 1. On the first day of Week 0, mice in the AOM/DSS, AOM/DSS + millet, and AOM/DSS + rice groups were intraperitoneally injected with AOM (10 mg/kg) in saline, while the NM group received the equivalent volume of saline. After the administration of a regular diet and water for a week, 2% DSS was added into the drinking water of the AOM/DSS, AOM/DSS + millet, and AOM/DSS + rice groups for 1 week followed by a 2 week recovery with intake of purified water. This cycle was repeated three times (the NM group was provided with drinking water without DSS). From Week 0 to the end of the study, mice in the NM and AOM/DSS groups were supplied with the AIN-93G rodent diet. For the AOM/DSS + millet and AOM/DSS + rice groups, the mice were fed with a modified AIN-93G rodent diet, in which corn starch was replaced by millet flour and rice flour, respectively. To correlate the content of dietary fiber and tryptophan in grains and the beneficial effects (e.g., activation of gut receptors), after analyzing the nutritional compositions of the grain flours, the original addition of dietary fiber was reduced from 6% to 3%. Thus, the total content of dietary fiber was same for the AOM/DSS and AOM/DSS + millet groups, and was higher than that of the AOM/DSS + rice group. The total content of tryptophan was similar between the AOM/DSS and AOM/DSS + rice groups and was lower than that of the AOM/DSS + millet group (Table 1).

The body weight and food consumption of each group were measured every other day throughout the experiment. The general appearance, signs of diarrhea, rectal bleeding, prolapse, and survival rates of the mice were inspected and recorded every week. The disease activity index (DAI) was calculated according to a previously described method, based on a combination of body weight loss, stool consistency and rectal bleeding [24]. The feces were collected in SPF environment one day before the euthanasia of the mice. All fecal samples were collected into sterile cryotubes immediately following defecation. During the study, mice that presented with more than 20% body weight loss, hunched posture, and a limited movement were euthanized to avoid excessive discomfort following the institutional animal care committee. At the end of the study, all mice were sacrificed by cervical dislocation after overnight fasting, and their serum and colon tissues were collected. In the AOM/DSS + rice group, one mouse which met the criteria for euthanasia two days before the end of the study was sacrificed and included into the final samples. All tissues and fecal samples were immediately frozen via liquid nitrogen, and then stored at −80 °C until further use.

### 2.4. Analysis of the Principal Nutritional Composition in Cereals

In this study, duplicate samples were determined according to the procedures of the Chinese national standards for moisture content (GB5009.3-2016), ash (GB5009.4-2016), protein content (GB5009.5-2016), fat content (GB5009.6-2016), and amino acid content (GB5009.124-2016). The total starch and dietary fiber (including soluble and insoluble dietary fiber) concentrations of cereals were measured by using Megazyme Test Kits (respectively based on AOAC Method 996.11 and AOAC Method 991.43). Analysis of the phenolics in the cereal was performed according to the Folin–Ciocalteu colorimetric method, following methods published previously with slight modifications [25].

### 2.5. Evaluation of Colonic Tumors

The evaluation of the colonic tumors was performed according to the methods previously described [26]. The colons were cut longitudinally, the feces were removed from the colon cavity, and the colon tissue was flushed by sterile saline. A general examination was then performed for megascopic colon tumors, including the tumor number and burden. Tumors were classified by the average diameter and enumerated: Small tumor (d < 3 mm), large tumor (d > 3 mm). Besides, the long diameter (L) and short diameter (S) of the tumors were measured with Vernier calipers and the tumor burden was calculated by the following formula: tumor volume (mm^3^) = L × S^2^/2. The tumor evaluation was performed with the help of a pathologist in a blinded manner. After general evaluation, the distal colon tissues (5 mm) were harvested and fixed in 10% neutral-buffered formalin for histological analyses. The remaining colon tissues were cut in half lengthwise and was wholly homogenized for mRNA and protein analyses.

### 2.6. Measurement of Cytokines

The levels of IL-1β, IL-10, IFN-γ, TNF-α, C-peptide (C-P), and lipopolysaccharides (LPS) in the serum were measured using commercially available kits (Sinouk Institute of Biological Technology, Beijing, China). The levels of cyclooxygenase-2 (COX-2) and prostaglandin E2 (PGE2) in the colon tissues were determined with commercial kits (USCN KIT INC, Wuhan, China), the monocyte chemotactic protein 1 (MCP-1) level was measured using an ELISA kit (LIANKE BIOTECH, CO., LTD, Hangzhou, China), and the myeloperoxidase (MPO) level was determined using an ELISA kit (Nanjing Jiancheng Bioengineering Institute, Nanjing, China).

### 2.7. Histopathological Evaluation

After being fixed in 10% formalin solution for 24 h, the distal colon tissues (5 mm) were paraffin-embedded, sectioned and stained using hematoxylin and eosin (H&E) [27]. Three random fields of views were photographed at 100× magnification with a microscope for each colon sample. The severity of inflammation was evaluated using histological scores based on the severity of crypt depletion and distortion (0–3), the degree of inflammatory infiltration (0–4) and the area of involvement (0–4) [28]. These evaluations were performed blind by a pathologist.

### 2.8. Determination of the SCFAs and Tryptophan Metabolites in Feces

The determination of SCFA levels in feces was performed according to our previously described method [29]. Before analysis, 50 mg of frozen feces was mixed with 0.45 mL 50% methanol (containing 0.1% formic acid) followed by cutting up and vortexing the samples. The concentrations of tryptophan metabolites were separated and analyzed using a Waters iclass-AB sciex 6500 liquid chromatography-tandem mass spectrometer (LC–MS/MS) equipped with a UPLC HSS T3 capillary column (2.1 cm × 50 mm i.d., 1.8 μm, Waters Co.). The optimized MS parameters for tryptophan metabolites were Appendix A. Data handling was carried out with an Agilent’s MSD ChemStation (E.02.00.493, Agilent Technologies, Inc., USA) and a MultiQuant (SCIEX, Framingham, MA).

### 2.9. Western Blot Analyses

Western blot analyses were conducted according to previously published procedures [30]. The primary antibodies against PCNA (1:1000), *p*-STAT3 (1:1000), STAT3 (1:1000), and BCL2 (1:1000), and the secondary antibody goat anti-rabbit IgG-HRP (1:5000) were polyclonal antibodies purchased from Wanleibio (Shenyang, China). The protein bands were developed using enhanced chemiluminescence.

### 2.10. Quantitative RT-qPCR

Quantitative RT-qPCR was performed following our previously published method [31]. Briefly, the total RNA was extracted from homogenized colon tissues using a TRIzol^TM^ Reagent (Thermo Fisher Sientific, China) according to the manufacturer’s instructions. The RNA concentration was determined using a nanophotometer (Implen, Germany) at a 260/280 nm absorbance ratio. Subsequently, 1 μg RNA that had been diluted to the appropriate range was subjected to reverse transcription with a LunaScript^TM^ SuperMix Kit (New England BioLabs, Ipswich, UK). SYBR qPCR Master Mix (ChamQ^TM^ Universal) was used for amplification, and specific DNA sequences were amplified with a Bio-Rad CFX ConnectTM Real-Time System (BIO-RAD, Hercules, CA, USA). The primers used are displayed in Appendix A. Genes encoding β-actin and 18 S were used as housekeeping genes (HSKGs). The HSKG with more similar PCR efficiency was used for the quantification of the target genes. The results of mRNA expression were calculated according to Pfaffl’s rule [32]. The relative expression ratio (R) of a target gene was calculated based on the efficiency (E) and the Ct deviation of the NM samples vs. the treatment samples and expressed in comparison to a reference gene:$$\mathrm{Ratio}=/{\left(E_{\mathrm{ref}}\right)}^{\Delta Ct_{\mathrm{ref}}\left(\mathrm{NM}-\mathrm{treatment}\right)}.$$

### 2.11. High-Throughput Sequencing and Bioinformatic Analysis

The extraction of DNA and the high-throughput sequencing were performed according to a previously described method [33]. The V3–V4 hypervariable regions of the bacterial 16 S rRNA were amplified with primers 338 F and 806 R. The meta-genomic sequencing was performed according to the standard protocols of Majorbio Bio-Pharm Technology Co. Ltd. (Shanghai, China). Additionally, the raw reads were deposited into the NCBI Sequence Read Archive (SRA), with the accession number SRP239677.

The bioinformatic analysis was performed using the I-sanger platform (Majorbio Bio-Pharm Technology Co. Ltd. Shanghai, China. www.i-sanger.com). The taxonomy of each 16 S rRNA gene sequence was analyzed by an RDP Classifier algorithm (http://rdp.cme.msu.edu/) against the Silva (SSU127) 16 S rRNA database using a confidence threshold of 70%. Refraction analysis and alpha-diversity analysis were performed using Mothur b.1.30.1. The rarefaction curves were used to confirm that the sequencing coverage was sufficient. Alpha-diversity was estimated through Sobs index and Shannon index. A heatmap based on the relative abundance of genera was generated using the R package 2.15. Weighted principal coordinate analysis (PCoA) was performed using R package, and statistical analysis was performed based on the values of PC1 with Unweighted-Unifrac distance. A hierarchical clustering analysis based on Unweighted-Unifrac distance was performed to evaluate the cage-effects caused by coprophagy of the mice maintained within the same cage. At least one sample from each cage was used to partially prevent a cage effect in the data.

### 2.12. Statistical Analysis

SPSS 21.0 was employed to perform the statistical analysis, and all the results are expressed herein as the mean ± standard deviation. One-way ANOVA was used to assess the significant differences (*p* < 0.05) between the means of each group. Two-way ANOVA was used when time was considered as a variable. Tukey test was performed as the post-hoc analysis in both the one-way and two-way ANOVA. Log-rank (Mantel–Cox) test was used to assess the significant difference for survival rate. For the bioinformatic analysis, the Kruskal–Wallis with Dunn’s test was used to assess the significant differences between groups. The data that were not normally distributed or did not display equal variance were logarithmically transformed to meet the criteria. All results were considered statistically difference at *p* < 0.05.

## 3. Results

### 3.1. Foxtail Millet Suppressed the Development of Colorectal Cancer in AOM/DSS-Treated Mice

During this experiment, the body weights of the NM group mice increased steadily. Mice in the AOM/DSS, AOM/DSS + millet, AOM/DSS + rice groups gained weight during the first 4 weeks. A loss of weight along with diarrhea and bloody stool began to be observed after the second DSS treatment, along with an increase in the DAI. During the following two week recovery period, the body weight loss and DAI gradually improved. After 9 weeks of dietary intervention, the survival ratio of the AOM/DSS + millet group was less than that of the NM group, but was improved compared to the AOM/DSS + rice group (Figure 2A). By the end of the study, the body weights of the mice supplied with millet were significantly higher than those of mice in the AOM/DSS + rice group, and no significant difference was found between the AOM/DSS and the AOM/DSS + rice groups (Figure 2B). The average food consumption of mice was shown in Appendix A. The DAI values were significantly reduced from Week 5 to the end of the study, in the millet-treated group compared to the AOM/DSS + rice group (Figure 2C). By the end of the treatment, a significant difference in the DAI was not observed between the AOM/DSS + millet mice and the AOM/DSS mice, but a significant reduction in the indexes of the millet-treated mice was observed during the first recovery period and the second treatment of DSS (from Week 2 to Week 5). In addition, all the mice that received the AOM/DSS treatment developed tumors. The number of tumors (<3mm) in the AOM/DSS group was significantly higher than that in the NM group and was reduced in the AOM/DSS + millet group (Figure 2D). The number of tumors (both <3 mm and >3 mm) was significantly lower in the millet-treated group than in the rice-treated group, and the tumor burden was suppressed in the AOM/DSS + millet group compared to the AOM/DSS and AOM/DSS + rice groups. These data indicate that, compared to the AOM/DSS + rice group, the millet diet ameliorated colonic inflammation and suppressed the formation of colitis-associated colonic tumors.

### 3.2. Principal Composition of Cereals

The principal components of millet, rice, and corn starch are shown in Table 2. The level of dietary fiber was determined to be 8.15% in millet, which was 1.6-fold of that in rice. Moreover, the contents of protein and fat in millet were higher than those in rice, but the content of starch was lower. Notably, the contents of tryptophan and total phenolics in millet were almost 3 times higher than those in rice.

The compositions of the modified AIN-93G rodent diets are shown in Table 1. The content of dietary fiber in AOM/DSS + millet group diet (6%) was the same as that in the standard diet (the NM and AOM/DSS groups) and was 1.2% greater than that in the AOM/DSS + rice group. The contents of tryptophan in the AOM/DSS + rice diet and the standard diet were similar, and the content of tryptophan in the AOM/DSS + millet diet was nearly 1.35-fold of that in the former two diets. The AOM/DSS + millet diet contained the most abundant phenolics and the lowest amount of starch compared to the standard and AOM/DSS + rice diets, and the components of protein and fat in the AOM/DSS + millet group diet were higher than those in the other three groups.

### 3.3. Foxtail Millet Attenuated Inflammation and Histological Pathology in the AOM/DSS-Treated Mice

The development of CRC was accompanied by aggravated colitis. To evaluate the effects of foxtail millet on colonic inflammation, we determined the levels of MPO, MCP-1, C-peptide, IFN-γ, and IL-1β (Figure 3A–E), which can be classified as key markers of inflammation. AOM/DSS treatment induced significant increases of the MPO and MCP-1 levels in colon tissues, suggesting an increase in neutrophil and macrophage recruitment, respectively. Compared to those in the AOM/DSS and AOM/DSS + rice groups, the mice treated with millet showed decreased levels of colon MPO and MCP-1. Moreover, the levels of serum C-P and IFN-γ in the AOM/DSS + millet group mice were significantly lower than those in the AOM/DSS + rice group, but no apparent difference was found in the level of serum IL-1β. The levels of C-P and IFN-γ in the AOM/DSS + rice group were significantly higher than those in the AOM/DSS group. Moreover, the concentration of serum LPS (Figure 3F), which induces inflammation, was significantly higher in the AOM/DSS group than in the NM group. The administration of millet significantly reduced the level of LPS compared to the AOM/DSS and AOM/DSS + rice groups.

The effects of AOM/DSS treatment and dietary intervention on colonic morphology were observed using histopathological analysis. Representative H&E-stained histological sections are shown in Figure 3G. The colons of the NM group exhibited a normal pathological morphology without inflammatory damage. In the AOM/DSS-treated mice, signs of inflammatory infiltration, depletion, and distortion of the crypts were observed in the colonic sections. Moreover, the severity of pathological damage, inflammatory infiltration, and distortion of the crypts were alleviated by the administration of millet compared to the AOM/DSS and AOM/DSS + rice groups.

### 3.4. Millet Regulated the Expression of Cytokines and Genes Involved in Colitis-Related Signaling Pathways

To explore the underlying molecular mechanisms, we determined the effects of foxtail millet on the expression of the key cytokines involved in the colitis-related signaling pathway. Compared to the NM group, AOM/DSS treatment significantly increased the colonic mRNA level of the pro-inflammatory cytokines TNF-α, IL-6, IL-17, COX-2 and iNOS (Figure 4). Compared to the AOM/DSS group, the mRNA expression levels of IL-6, IL-17, COX-2, and iNOS in the mice treated with millet were downregulated, while the pro-inflammatory cytokines tested were significantly upregulated in the AOM/DSS + rice-treated mice versus those in the AOM/DSS + millet group. Moreover, the mRNA expression level of FOXP3, a key regulatory factor in colitis, was significantly elevated in the mice of the AOM/DSS + millet group compared to those of both the AOM/DSS and AOM/DSS + rice groups. Although there was no apparent difference in the expression of anti-inflammatory cytokine IL-10 between the AOM/DSS group and the AOM/DSS + millet group, the mRNA expression of IL-10 was significantly reduced in the AOM/DSS + rice group. Moreover, compared to the AOM/DSS and AOM/DSS + rice groups, millet administration significantly promoted the expression of IL-22 significantly, which is beneficial to intestinal integrity. Zonulae occludens-1 (ZO-1) and occludin play important roles in maintaining intestinal integrity. The mRNA expression of ZO-1 and occludin was significantly decreased by AOM/DSS administration compared to levels in the NM group. The dietary intake of millet significantly upregulated the mRNA expression of ZO-1 and occludin compared to levels in the AOM/DSS and AOM/DSS + rice groups.

These RT-qPCR findings were further confirmed via ELISA. There was no significant difference in the concentration of serum TNF-α among the NM, AOM/DSS and AOM/DSS + millet groups. However, compared to levels in the AOM/DSS + millet group, the intake of rice significantly enhanced the concentration of serum TNF-α. The concentration of anti-inflammatory cytokine IL-10 in the serum was significantly increased by millet treatment compared to that in the AOM/DSS group. Moreover, the concentrations of the pro-inflammatory cytokines COX-2 and PGE2 in colon tissues were significantly upregulated by AOM/DSS treatment. The dietary intervention of millet significantly reduced the levels of colonic COX-2 and PGE2 compared to the levels for the AOM/DSS- and AOM/DSS + rice-treated mice.

Taken together, foxtail millet demonstrated the ability to regulate the production of key cytokines involved in the colitis-related signaling pathway and improve intestinal integrity compared to rice.

### 3.5. Inhibition of the Development of Colorectal Cancer by Foxtail Millet

To understand the molecular mechanisms underlying the impact of colonic tumorigenesis, we determined the effects of millet treatment on the key molecules related to the development of CRC; the results are shown in Figure 5A. The phosphorylation of STAT3, which plays a key role in regulating cell growth, proliferation, and survival, was significantly activated by exposure to AOM/DSS. The millet-treated mice showed significantly inhibited phosphorylation of STAT3 compared to the AOM/DSS mice and the rice-treated mice. Furthermore, the expression of Bcl-2 (an anti-apoptosis factor) and PCNA (a cell-proliferation factor) in colon tissues was increased in the AOM/DSS group compared to the NM group. The dietary intake of millet significantly inhibited the expression of Bcl-2 and PCNA compared with the AOM/DSS and the AOM/DSS + rice groups. Moreover, the mRNA expression of VEGF (vascular endothelial growth factor), which is a sign of angiogenesis, was significantly downregulated in the millet-treated mice compared to the AOM/DSS mice and the rice-treated mice (Figure 5B).

### 3.6. Microbial Metabolites of Foxtail Millet Activated AHR and GPCRs

Compared to rice, foxtail millet contains more abundant tryptophan and dietary fiber, which have been shown to activate gut receptors and promote gut health after being metabolized by the gut microbiota. To explore the mechanism through which millet impacts colon health, we measured the fecal contents of tryptophan and dietary fiber metabolites and determined the mRNA expression levels of specific gut receptors.

The changes in fecal SCFAs in different groups are shown in Figure 6A. Compared to the NM group, the fecal concentrations of acetic acid, propionic acid, and butyric acid were significantly decreased by AOM/DSS treatment. The results were consistent with previous studies [34], and was supposed to be induced by the reduced abundance of SCFA-producing bacteria. The intake of millet significantly increased the contents of acetic acid, propionic acid and butyric acid relative to the AOM/DSS + rice group. No difference in the contents of acetic acid and butyric acid was observed between the AOM/DSS and the AOM/DSS + millet groups.

Next, we determined the contents of fecal tryptophan metabolites. Among the determined tryptophan metabolites, indole-3-propionic acid (IPA), 3-methylindole (3 ML), indole lactate (ILA), indole acetic acid (IAA), and indole acrylic acid (IA) are specific ligands of AHR. IPA, IAA, and IA contents were significantly reduced by exposure to AOM/DSS compared to those in the NM group (Figure 6B). Compared to the AOM/DSS group and the AOM/DSS + rice group, millet significantly increased the production of IPA, IAA, and IA. Moreover, the total contents of tryptophan metabolites with AHR-activation activity were increased in the AOM/DSS + millet group. The fecal content of other tryptophan metabolites is shown in Appendix A.

We then determined the mRNA levels of AHR, GPCR41, and GPCR43 (Figure 6C–E). Consistent with the decreased production of tryptophan metabolites and SCFAs, AOM/DSS treatment downregulated the mRNA expression levels of AHR, GPCR43, and GPCR41 relative to the NM group. The AOM/DSS + millet group showed a significantly upregulated expression of AHR compared to the AOM/DSS group and rice-treated group. Meanwhile, the mRNA expression of two SCFA receptors was significantly upregulated in the millet-treated mice compared to the rice-treated mice. These results suggest that the dietary intake of foxtail millet could increase the production of tryptophan metabolites and SCFAs and activate the related gut receptors.

### 3.7. Effects of Foxtail Millet on the Regulation of Gut Microbiota

A Miseq analysis of the bacterial 16 S rRNA in fecal samples was used to explore the effects of millet and rice on the composition of the gut microbiome (Figure 7). Compared to the NM group, the Shannon index and Sobs index in the AOM/DSS group were significantly reduced (Figure 7A,B), indicating that the α-diversities decreased in the AOM/DSS-induced mice. The dietary intake of millet significantly upregulated the α-diversities compared to the AOM/DSS group. No significant difference in the α-diversities was found when comparing the AOM/DSS + rice group with the AOM/DSS group or the AOM/DSS + millet group. The alteration of the gut microbiome’s composition was examined via principal coordinate analysis (PCoA) at the OTU level (Figure 7C). The distance between the samples of the three AOM/DSS-treated groups and those in the NM group was apparent, suggesting that AOM/DSS significantly changed the structure of the gut microbiome. The intervention of millet regulated the composition of gut microbiome toward the NM group.

Relative abundance of microbial phylotypes was compared between different groups (Figure 8). At genus level, the abundance of norank_f_*Bacteroidales_S24-7*, *Parabacteroides*, *Parasutterella*, *Alloprevotella*, *Ruminiclostridium_9*, *Odoribacter*, norank_f_*Lachnospiraceae*, *Rikenella*, *Anaerotrunces*, and *Oscilibacter* was significantly decreased, and the abundance of *Escherichia-shigella* and *Erysipelatoclostridium* was significantly increased in the AOM/DSS group compared to the NM group. Dietary consumption of millet significantly increased the abundance of *Allobaculum*, *Bifidobacterium* and norank_f_*Bacteroidales_S24-7* and decreased the abundance of *Alistipes* in compared to the AOM/DSS + rice group.

In addition, the difference of the value of PC1 between different cages within a single group was analyzed (Appendix A) and a hierarchical clustering analysis (Appendix A) was performed to evaluate the cage-effects caused by coprophagy of the mice maintained within the same cage.

## 4. Discussion

Epidemiological studies have suggested that high wholegrain or coarse cereal intake is associated with a reduced risk of CRC [15]. However, there is still a lack of strong evidence from intervention studies showing whether intake of certain cereals can reduce the risk of CRC. Moreover, the molecular mechanisms by which these grains regulate intestinal health, thus impacting the progression of CRC, remain largely unexplored. In the present study, we compared the effects of foxtail millet and rice on colitis-associated CRC. Compared to rice, foxtail millet exhibited better effects in ameliorating colonic inflammation and inhibiting the progression of AOM/DSS-induced colitis-associated CRC. This was supported by the reduced levels of DAI, numbers of tumor, overall tumor burden, and the improved survival rate and histological score in the millet-treated mice compared to the rice-treated mice. Also, the consumption of millet decreased the level of tumor burden compared to the AOM/DSS mice, suggesting a beneficial effects of millet intake during the progression of CRC.

Chronic intestinal inflammation results in defects of the intestinal epithelium along with the dysregulation of intestinal mucosal immunity [35]. C-P is a key marker used to assess the severity of inflammation, and its concentration is positively correlated with the incidence of CRC [36]. IFN-γ, a cytokine elevated in IBD patients, is frequently used to assess colon inflammation [37]. MPO is a marker of the recruitment of neutrophils and is involved in producing inflammatory cytokines [38]. During the study, mice which met the criteria of euthanasia were sacrificed to avoid excessive discomfort following the guidelines of the institutional animal care committee. The sample size of the physiological parameters is the number of mice at the end of the study. In the present study, significant differences in these inflammatory biomarkers were found between the millet-treated mice and the rice-treated mice. It should be noticed that even though the rice-treated mice presented more serious symptoms than both the AOM/DSS and the AOM/DSS + millet mice, this does not mean that the intake of rice induced intestinal toxicity. Rice is the most commonly consumed refined grain in Asia. The results of the present study support the prior dietary recommendations based on millet or a millet-involved food matrix for IBD patients and the high-CRC-risk population. Particular attention should be paid to developing a rice-dominant recipe that focuses on dietary balance.

Researchers have proven that pro-inflammatory cytokines including IL-6 and TNF-α can activate the downstream transcription factor STAT3 by binding to their soluble receptors, and subsequently upregulating the expression of genes related to cell survival, proliferation and growth such as Bcl-2, PCNA and VEGF, resulting in the progression of CRC [11,39]. COX-2, a pro-inflammatory enzyme, converts free arachidonic acid into PGE2, which can upregulate the production of IL-6 [40]. IL-17 can stimulate the release of COX-2 and several inflammatory cytokines, including IL-6, TNF-α, and IFN-γ [41,42]. FOXP3, a transcriptional factor expressed in Treg cells, can enhance the production of anti-inflammatory cytokine IL-10 and inhibit the production of IL-17 [43,44]. In the present study, compared to rice, supplementation with millet significantly increased the levels of FOXP3 and reduced the levels of colonic pro-inflammatory cytokines IL-17, IL-6, and TNF-α. Accordingly, decreased levels of COX-2 and PGE2, as well as decreased levels of STAT3 phosphorylation, were observed in the millet-treated mice. The levels of Bcl-2, PCNA, and VEGF were reduced after millet treatment. These results suggest that the chemoprotective effects of a millet-rich diet are mediated by the cytokine/STAT3 pathway. However, for the immunoblotting and RT-qPCR analyses, the colon samples were totally collected only at the end of the study. The abundance of cancer cells may influence the expression of proteins which are highly expressed in the colon tissues (e.g., VEGF and PCNA). Thus, it is difficult to determine whether the decreased expression of these proteins are directly induced by the treatment of millet. Further studies on the dynamic processes of the development of CRC are warranted to clarify the mechanism of the beneficial effects of diet rich in millet.

We further speculate that differences in the nutritional components of millet and rice may explain their effects on the progression of CRC. We found that foxtail millet contains more abundant tryptophan and dietary fiber than rice. Undigested and unabsorbed tryptophan and dietary fiber both reach the colon and their microbial metabolites have been reported to have beneficial effects on colon health [45]. Tryptophan is an essential amino acid for which humans rely on dietary supplementation, and can be converted into indole and indole derivatives by the gut microbiota [9]. These metabolites are specific ligands of AHR, which have a considerable effect on maintaining intestinal homeostasis. Intestinal bacteria can ferment dietary fiber into SCFAs, which act as natural ligands for GPCR41 and GPCR43. In the present study, the intake of millet enhanced the production of tryptophan metabolites and SCFAs. Accordingly, the gene expression levels of AHR and GPCRs were significantly upregulated in the millet-treated mice. Among the tested tryptophan metabolites, IPA, 3 ML, ILA, IAA, and IA have been reported as specific ligands of AHR [8,9]. The results suggest that the metabolism of dietary components by the gut microbiota play an important role in the chemoprotective effects of a millet- rich diet. The activation of AHR and GPCRs could upregulate the expression of FOXP3, IL-10, and IL-22, which work together to promote the secretion of colonic mucins [8,46]. Moreover, AHR and GPCRs activations contribute to the downregulation of anti-inflammatory cytokines, including TNF-α and IL-6 [8,10]. Taken together, the inhibited expression of inflammatory cytokines and the upregulated expression of occludin and ZO-1 in the millet-treated mice have been mediated by the activation of gut receptors. Meanwhile, these beneficial effects contributed to the chemoprotective effects of foxtail millet on early-stage CRC. A potential molecular mechanism is shown in Figure 9.

The contribution of gut receptors to the prevention of CRC can also be verified by comparing the differences between the AOM/DSS mice and the rice-treated mice or millet-treated mice. On the one hand, the content of total dietary fiber was higher in the AIN-93G diet than in the rice diet, while the content of tryptophan was not very different. Consequently, the levels of fecal SCFA and colonic GPCR mRNA expression in the AOM/DSS mice were higher than those in the rice-treated mice, while the levels of fecal tryptophan metabolites and colon AHR mRNA expression showed no difference. Accordingly, some of the down-stream biomarkers associated with inflammation and tumorigenesis were increased in the rice-treated mice compared to the AOM/DSS mice. On the other hand, the content of dietary fiber in the AIN-93G diet was the same as that in the millet diet, while the millet diet had a higher content of tryptophan. Accordingly, the levels of fecal tryptophan and AHR mRNA expression were increased in the millet-treated mice and the inflammation associated biomarkers were further down-regulated. This evidence supports the assertion that both AHR and GPCRs play a role in the aforementioned beneficial effects.

Metagenomic studies have suggested that intestinal flora disorders are key risk factors for inflammatory bowel disease and CRC [47]. Microbes such as *Bacteroides*, *Alistipes,* and *Escherichia-shigella* were observed to increase in CRC patients [14] and have been confirmed to promote the formation of colorectal tumor [48,49]. Probiotics such as *Bifidobacterium breve* can alleviate CRC by inhibiting the inflammatory reactions and DNA methylation of the host, thereby inducing colitis and polyp retrogression, and activating tumor suppressor gene expression. Furthermore, a decline in microbial diversity, which is a sign of microbial dysbiosis, is considered to be an important factor for the incidence of CRC [34,37]. In this study, the administration of AOM/DSS reduced the microbial diversity and abundance of several SCFA-producing bacteria, including norank_f_*Bacteroidales_S24-7*, *Parabacteroides* and *Alloprevotella*. This change may result in the decreased levels of SCFAs in the AOM/DSS-treated mice. Compared to the rice-treated group, the consumption of millet improved the relative abundance of *Bifidobacterium*, which is considered to be probiotics [45,50]. Meanwhile, millet supplementation decreased the abundance of pathogenic bacteria *Alistipes*. Norank_f_*Bacteroidales_S24-7*, which was demonstrated to deplete during colonic inflammatory phases [51] and contains several butyrate-producing bacteria, was improved by millet consumption in the current study. These results suggest that a change in the gut microbiome played a role in ameliorating carcinogenesis in the millet-treated mice. However, only 28 OTUs out of the 585 detected OTUs were found to be significantly changed between the gut microbiota of the millet-treated mice and the rice-treated mice. This may be because the types of components are similar between the two grains. Besides, mice are coprophagic and may share microbiomes within a cage. The statistical method for the microbiome analysis should employ a mixed model with cage as a nested factor. In the present study, the sample size within a single cage did not met the criteria of the above-mentioned statistical analysis. Thus, further studies are needed to increase the sample size and to clarify how significantly the modulation of grains on gut microbiota composition contributes to intervening in the progression of CRC.

## 5. Conclusions

In this study, the effects of the dietary intake of millet and rice on the progression of CRC were compared for the first time. The current findings suggested that the intake of foxtail millet attenuated colonic inflammation and reduced the risk of AOM/DSS-induced colitis-associated CRC in mice. The regulatory effects were mediated by the activation of AHR and GPCRs and the inhibition of STAT3 phosphorylation by the microbial metabolites of the foxtail millet. The present study provides useful information for building a better dietary pattern for IBD and early-stage CRC patients. Furthermore, it provides intervention evidence for the beneficial effects of coarse grains and whole grains. Further studies are still needed to discuss the chemo-protective effects of different kinds of grains on the progression of CRC in animal models and clinical trials, and to identify the bioactive components in these grains.

## Figures and Tables

**Figure 1 nutrients-12-02367-f001:**
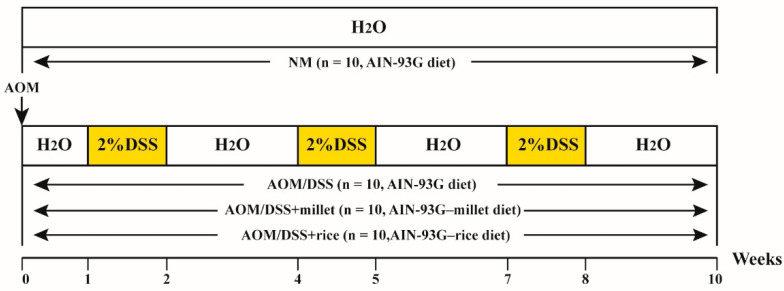
Experimental protocol for the establishment of a murine colitis-associated cancer (CAC) model induced by azoxymethane/dextran sodium sulfate (AOM/DSS).

**Figure 2 nutrients-12-02367-f002:**
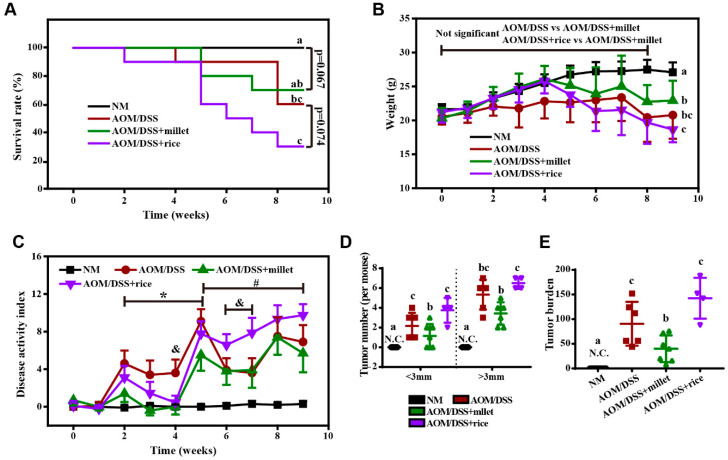
General observation of the mice in NM, AOM/DSS, AOM/DSS + millet and AOM/DSS + rice groups. (**A**) Survival rate; (**B**) Weight; (**C**) Disease activity index (DAI); (**D**) Tumor number (per mouse); (**E**) Tumor burden (total tumor volume per mouse in mm^3^). (**A**) Survival rate was analyzed by Log-rank (Mantel–Cox) test. (**B**,**C**) Two-way ANOVA followed by Tukey post hoc test (*n* = 4–10 at the last timepoint) was used to analyze the significance at each time point during the whole period of the study. The brackets indicate the comparison between groups at certain time points. For Figure 2C, * *p* < 0.05 for AOM/DSS vs. AOM/DSS + millet; # *p*< 0.05 for AOM/DSS + millet vs. AOM/DSS + rice; & *p* < 0.05 for AOM/DSS vs. AOM/DSS + rice at. (**D**,**E**) One-way ANOVA followed by Tukey post-hoc test. Data are shown as the mean ± SD. Means with different letters are significantly different (*p* < 0.05). *n* = 4–10/group. NM, normal.

**Figure 3 nutrients-12-02367-f003:**
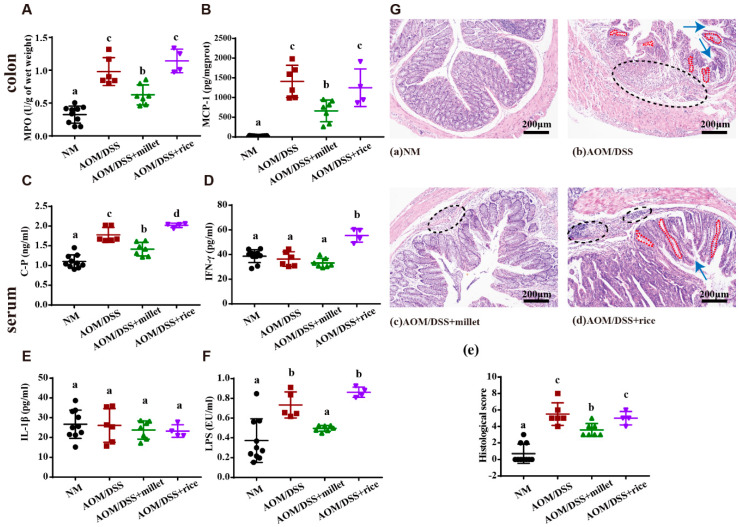
Macroscopic and histological assessments of colitis. Colonic (**A**) myeloperoxidase (MPO) and (**B**) monocyte chemotactic protein 1 (MCP-1); serum (**C**) C-P, (**D**) IFN-γ, (**E**) IL-1β, and (**F**) LPS. (**G**) Representative H&E staining of the formalin fixed sections of the colonic tissues. The black ellipses indicate inflammatory infiltration; the blue arrows indicate crypt defect; and the red curves indicate crypt distortion. Magnification: ×100 (upper). Data are shown as the mean ± SD. Means with different letters are significantly different (*p* < 0.05). *n* = 10, 6, 7, and 4 for NM, AOM/DSS, AOM/DSS + millet, and AOM/DSS + rice groups, respectively.

**Figure 4 nutrients-12-02367-f004:**
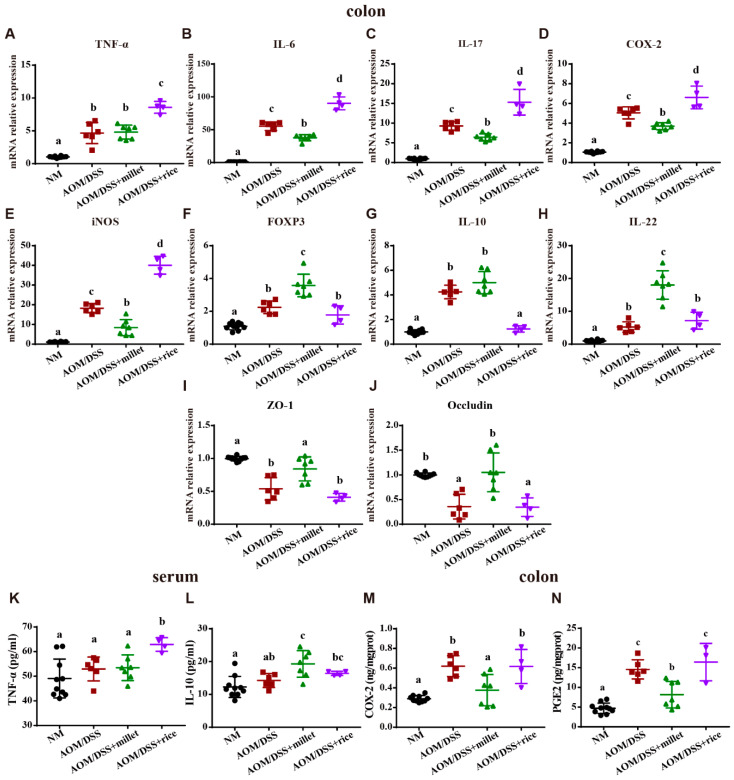
The mRNA expression levels of (**A**) TNF-α, (**B**) IL-6, (**C**) IL-17, (**D**) COX-2, (**E**) iNOS, (**F**) FOXP3, (**G**) IL-10, (**H**) IL-22, (**I**) ZO-1, and (**J**) occludin in the colonic samples. The expression levels of (**K**) TNF-α and (**L**) IL-10 in serum. The expression levels of (**M**) COX-2 and (**N**) PGE2 in the colon. Data are shown as the mean ± SD. Means with different letters are significantly different (*p* < 0.05). *n* = 10, 6, 7, and 4 for NM, AOM/DSS, AOM/DSS + millet, and AOM/DSS + rice groups, respectively.

**Figure 5 nutrients-12-02367-f005:**
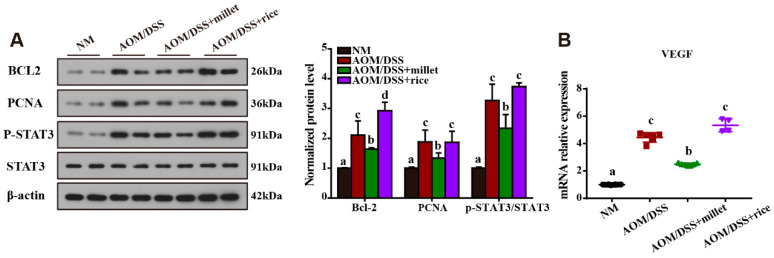
(**A**) Western blotting of B-cell lymphoma-2 (Bcl-2), proliferating cell nuclear antigen (PCNA), P-STAT3, and signal transducer and activator of transcription 3 (STAT3) in the colonic samples. (**B**) mRNA expression of vascular endothelial growth factor (VEGF) in the colon. Data are shown as the mean ± SD. Means with different letters are significantly different (*p* < 0.05). *n* = 10, 6, 7, and 4 for NM, AOM/DSS, AOM/DSS + millet, and AOM/DSS + rice groups, respectively.

**Figure 6 nutrients-12-02367-f006:**
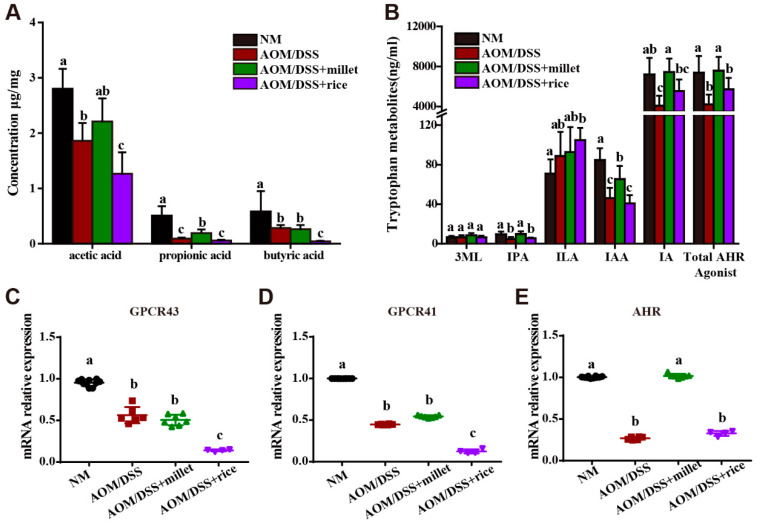
The concentrations of SCFAs (**A**) and tryptophan metabolites in feces (**B**). The mRNA expression levels of GPCR43 (**C**), GPCR41 (**D**), and AHR (**E**) in the colon. Data are shown as the mean ± SD. Means with different letters are significantly different (*p* < 0.05). *n* = 10, 6, 7, and 4 for NM, AOM/DSS, AOM/DSS + millet, and AOM/DSS + rice groups, respectively.

**Figure 7 nutrients-12-02367-f007:**
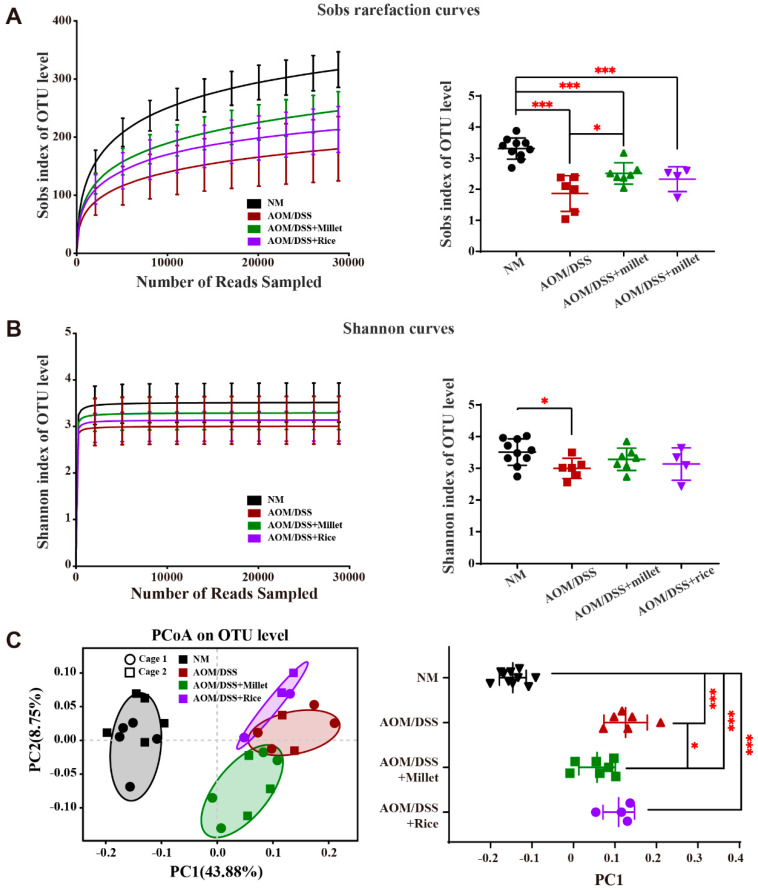
Effects of millet and rice on gut microbial diversity and composition. (**A**) Sobs index and rarefaction curves. (**B**) Shannon index and rarefaction curves. (**C**) Unweighted Unifrac PCoA analysis of gut microbiota metagenomes at the OTU level. * *p* < 0.05, *** *p* < 0.001 by Kruskal–Wallis with Dunn’s test. *n* = 10, 6, 7, and 4 for the NM, AOM/DSS, AOM/DSS + millet, and AOM/DSS + rice groups, respectively.

**Figure 8 nutrients-12-02367-f008:**
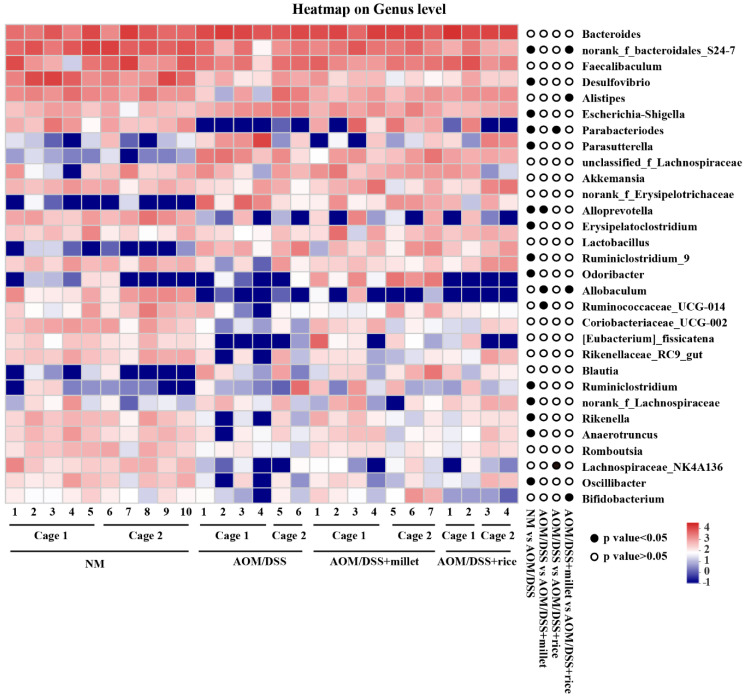
Heatmap analysis at genus level. ● *p* < 0.05 by Kruskal–Wallis with Dunn’s test. *n* = 10, 6, 7, and 4 for the NM, AOM/DSS, AOM/DSS + millet, and AOM/DSS + rice groups, respectively.

**Figure 9 nutrients-12-02367-f009:**
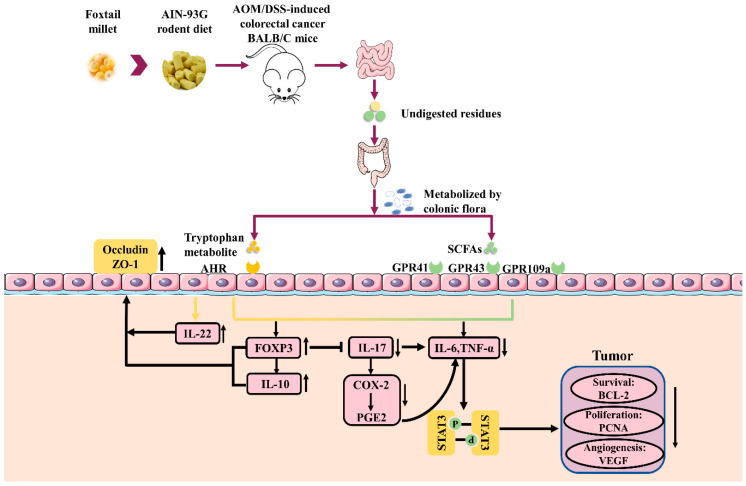
Mechanism of the regulatory effects of foxtail millet on the progression of colitis-associated colorectal cancer (CRC).

**Table 1 nutrients-12-02367-t001:** Composition of experimental diets.

	NM/AOM-DSS	AOM/DSS + Millet	AOM/DSS + Rice
**Ingredient (g/kg Diet)**			
Millet flour	-	397.0	-
Rice flour	-	-	397.0
Corn starch	397.0	-	-
Casein	200.0	200.0	200.0
L-Cystine	3.0	3.0	3.0
Maltodextrin	132.0	132.0	132.0
Sucrose	100.0	100.0	100.0
Soybean oil	70.0	70.0	70.0
Cellulose	60.0	30.0	30.0
Mineral mix (AIN-93G-MX)	35.0	35.0	35.0
Vitamin mix (AIN-93G-VX)	10.0	10.0	10.0
Choline Bitartrate	2.5	2.5	2.5
Butylhydroquinone	0.014	0.014	0.014
**Analyzed nutritional composition (g/kg diet)**			
Protein	200.0	236.0	223.0
Fat	70.0	85.0	73.0
Total starch	329.1	214.1	288.3
Total dietary fiber	60.0	60.0	48.0
Tryptophan	1.25	1.85	1.35
Total phenolics (mg GAE/100g)	0.0019	0.0282	0.0084

NM, normal; GAE, gallic acid equivalent; AOM/DSS, azoxymethane/dextran sodium sulfate.

**Table 2 nutrients-12-02367-t002:** Composition of millet, rice, and corn starch.

Composition	Content (g/100 g)		
	Foxtail Millet	Rice	Corn Starch
Moisture	9.58	13.00	13.54
Ash	1.16	0.40	0.17
Protein	10.51	7.20	1.32
Fat	4.32	1.13	0.42
Total starch	53.93	72.64	82.90
Total dietary fiber	8.15	5.14	0.58
Soluble dietary fiber	0.81	0.86	0.36
Insoluble dietary fiber	7.34	4.28	0.22
Tryptophan	0.18	0.06	0.03
Total phenolics (mg GAE/100 g)	39.05	11.17	4.73

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
