# Peer review of "Dietary Supplementation of Foxtail Millet Ameliorates Colitis-Associated Colorectal Cancer in Mice via Activation of Gut Receptors and Suppression of the STAT3 Pathway"

_nutrients, 2020, doi:10.3390/nu12082367_

Round 1

Reviewer 1 Report

Line 38 - reword.  Cancer is not "the premier" factor causing death worldwide. Leading causes of death worldwide are heart disease and stroke.  See https://www.who.int/news-room/fact-sheets/detail/the-top-10-causes-of-death

Consider working with an English writing service to correct numerous grammatical errors.

line 56 - Can you clarify which GPCR is expressed lower in IBD patients?  The acronym GPCR is non-specific.

The introduction does not provide adequate review of the current state of the science with respect to dietary cereal intake (grains or fiber) on colorectal cancer.  This is a well-studied field that needs for that two short sentences and a single citation of a general review article.

The AIN93G diet used for groups 1-3 had 60 g/kg fiber, while the diet with rice flour had just 48 g/kg fiber. Thus, it will be difficult to attribute observed differences between rice and millet diets groups to the food source as opposed to the fiber content.

How were the animals housed?  How many mice per experimental group?  Were the colon tissues blinded before determining tumors?  Were tumors verified by a board-certified pathologist?  How were colon tissues collected for protein and mRNA measurements?  Were those mucosal scrapings that included tumor tissue?

Again, for the histopathology evaluation of inflammation, were the samples blinded before visualization?  What grading scale was applied?  Who performed the evaluation?

For antibodies, clarify whether those were monoclonal or polyclonal.  Also, include details on secondary antibody and detection methods.

From the list of primers, it appears that you applied beta actin as your sole housekeeping gene, and used the delta-delta Ct method for quantitation.  This is a subpar approach to qPCR.  One, beta actin is typically expressed at much much higher levels than these other genes, making most PCR assays highly efficient.  The delta-delta Ct method for quantitation does not account for the differing PCR efficiencies for different assays.  Rather, employ multiple HSKGs (at minimum 2, preferably with amplification at near the same Ct values as your target genes) for normalization and use the Pfaffl method for quantitation after generating PCR standards for each assay.

For the microbiota sequencing, reference 24 does not provide thorough details.  Please include those details here or cite  the appropriate references directly with all the necessary details for someone to replicate your work, including processing the data. 

Statistical analyses may not be appropriate if the animals were group housed.  Mice are coprophagic and largely share microbiomes within a cage.  If they were group housed, your statistics should instead employ a mixed model with cage as a nested factor.  Moreover, you did not describe any data validation analyses, such as tests for normal distribution or equal variance, which are underlying assumptions with the ANOVA.  Were any transformations applied to the raw data?  Also, as I see results presented in figure 2, you clearly have data with two main factors:  time point and experimental diet.  As you are clearly looking at diet at different time points, then your statistical model must include both main factors (and their interaction).  You may need to consult with a statistician to reanalyze your data.

Figure 2A This plot infers that some mice were lost during the study. What were the criteria for euthanasia?

What statistical test was applied for survival analysis?  What is meant by survival "amount"?  Do you mean the number of animals?  If so, does that mean that by the end of the study, your sample size for the AOM/DSS+rice group was only 3?  

Please provide food intake data to aid in interpretation of the results.  

As of figure 2, we still do not know how many mice per treatment group.  

Tumor incidence is the number of mice with tumors (any number of tumors).  Please label Fig 2D correctly on the Y axis, as the values shown are multiplicity.  Also, did all mice exposed to AOM/DSS in this model develop tumors?  For panel D, the legend indicates that means with different letters are significantly different, however <3mm for AOM/DSS+millet does not appear different than > 3mm AOM/DSS+rice.  I suspect separate analyses were applied for different tumor sizes.  Again, here you have two factors:  tumor size and diet treatment.  Please clarify these results and apply the proper statistical test.

The authors should consider calculating tumor burden, the total tumor volume for each individual mouse.  Also, the authors seem to overlook the lack of difference for the AOM/DSS+millet compared to the control AOM/DSS group.  The statement line 204 might be an over-exaggeration given that large tumors seem unaffected, which would be concerning as these are likely the more aggressive or advanced tumors.

For H&E images, how many fields of view were examined for each colon?  Are these representative images?

For the protein and transcript analyses, how do you account for the presence of tumor tissue in those samples.  It seems to me that you have patterns of gene expression that likely reflect the tumor state, rather than earlier molecular events that lead to tumor development that were influenced by the diet.  With some colon tissues having higher tumor burden than other, how do you know that these apparent differences in expression are related to the dietary treatments rather than differences in the abundance of cancer cells in the colon samples?  VEGF and PCNA are good examples; one would rightly expect their expression to be highly correlated with tumor abundance.  This flaw is the main problem of animal studies designed this way, where samples are collected only at the end when disease is manifest, rather than collecting cohorts of samples throughout the study to understand dynamic processes leading to disease development.

Please plot rarefaction curves as means + error for easier visualization.  Also, please be consistent in coloring and graph style to match the prior plots.  That bright yellow is hard to see on a white background.  Also the legend is incomplete, in that the symbols are not explained.  Why no error bars for bar graphs for 7A and 7B?

Was alpha diversity for the millet diet group different than the rice group?

Statistical analyses appear to have been performed for the bacteria taxonomic data.  However, an ANOVA is not the right test, as these data are very likely to be zero-inflated with a left shifted distribution.  Consider using a tool like Microbiome Analyst to help work with these data. https://www.microbiomeanalyst.ca/faces/home.xhtml

Comparing proportions between only the millet vs. rice doesn't make much sense.

Discussion, line 358-360 reads as overly broad regarding the main endpoints.  At the end of the study, DAI was not different between the millet and rice diets.  And, while the millet diet did lead to fewer tumors (both small and large) compared to the rice diet, that was only the case for the small tumors compared to the standard diet.  It's not known whether the overall tumor burden was different.  Line 360 - tumor incidence data were not shown.  Overall, this first paragraph is an overstatement of these findings.

The discussion fails to address any limitations of this study, of which I have identified quite a few that make interpretation of the results quite challenging.

Author Response

Response to Reviewer 1 Comments

Dear Editors and Reviewers:

Thank you very much for your letter and comments on our manuscript entitled “Dietary supplementation of foxtail millet ameliorates colitis-associated colorectal cancer in mice by activation of gut receptors and suppression of STAT3 pathway” (ID: nutrients-869563). We greatly appreciate the constructive comments that are very helpful for the revision of our manuscript. We have made additions and corrections according to your comments. The revised sections in the manuscript are highlighted in red.

We do hope that the revised manuscript adequately addressed your comments. The major corrections and the responses to the reviewers’ comments are as follows:

Point 1: Line 38 - reword.  Cancer is not "the premier" factor causing death worldwide. Leading causes of death worldwide are heart disease and stroke.  See https://www.who.int/news-room/fact-sheets/detail/the-top-10-causes-of-death. 

Response 1: Thanks for your suggestion. According to another data from WHO, cancer is the second leading cause of death worldwide (https://www.who.int/en/news-room/fact-sheets/detail/cancer). The difference may be due to that whether the data considering the cancer population as a whole or dividing it into different types of cancers. In order to avoid misunderstanding, some descriptions about the causes of death were removed, and the revision was supplemented on Page 1, Line 38-39.

Point 2: Consider working with an English writing service to correct numerous grammatical errors.

Response 2: Thanks for your suggestion. The manuscript has been thoroughly revised by the MDPI English editing service, with a certificate number of english-20750.

Point 3: Line 56 - Can you clarify which GPCR is expressed lower in IBD patients?  The acronym GPCR is non-specific.

Response 3: GPCR43 was found to be lower expressed in IBD patients, and the activation of both GPCR41 and GPCR43 showed protective effects against IBD. The manuscript has been revised to clarify the specific GPCRs which are associated with IBD and the content of this study. The revision was supplemented on Page 2, Line 54 and Line 57.

Point 4: The introduction does not provide adequate review of the current state of the science with respect to dietary cereal intake (grains or fiber) on colorectal cancer.  This is a well-studied field that needs for that two short sentences and a single citation of a general review article.

Response 4: Thank you for your suggestion. The introduction has been revised, and the current state of the research on the effects of dietary cereal and fiber intake on colorectal cancer was reviewed on Page 2, Lines 78-80.

Point 5: The AIN93G diet used for groups 1-3 had 60 g/kg fiber, while the diet with rice flour had just 48 g/kg fiber. Thus, it will be difficult to attribute observed differences between rice and millet diets groups to the food source as opposed to the fiber content.

Response 5: Thanks for your comments. As you mentioned, the beneficial effects of millet may be due to the fiber content. In fact, this is a main conclusion of the present study. The beneficial effects of millet are expected to the balanced nutrition component. In the present study, we found that the microbiota metabolites of fiber and tryptophan, which belong to nutrition component, is an important factor that influences the effects of grains on the progression of CRC. The original purpose of the diet design was to further confirm the beneficial effects of fiber. The content of fiber was the same in the AOM/DSS and the AOM/DSS-millet diet, and was higher than that in the AOM/DSS-rice diet. Correspondingly, the levels of fecal SCFA and mRNA expressions of GPCRs were relatively similar in the AOM/DSS and the AOM/DSS-millet groups, and were both higher than that in the AOM/DSS-rice group. Thus, from the comparison between the mRNA expressions of gut receptors and the content of fiber, the effects of fiber and the corresponding signaling pathways involved could be further confirmed. We have revised the manuscript to clarify the reason of the diet design on Page 3, Lines 125-131.

Point 6: How were the animals housed?  How many mice per experimental group?  Were the colon tissues blinded before determining tumors?  Were tumors verified by a board-certified pathologist?  How were colon tissues collected for protein and mRNA measurements?  Were those mucosal scrapings that included tumor tissue?

Response 6: Thanks for your comment. The mice were housed in groups of five mice per cage, and were maintained in a room isolated from all other ongoing animal experiments and were handled only by the primary investigators. This information was supplemented on Page 3, Lines 103-105.

The colonic tumor was evaluated in a blind manner and was verified with the help of a colleague who is a certified doctor (certificate No: 201712110120106199107080015) majoring in pathology. The distal colon tissues (5 mm) were used for histological analyses. The remaining colon tissues were cut in half lengthwise and was wholly homogenized before mRNA and protein analyses. The protocol of colon tissues collection was performed according to a previous study (Sougiannis, A.T., et al. Impact of 5 fluorouracil chemotherapy on gut inflammation, functional parameters, and gut microbiota. Brain Behav Immun 2019, 80, 44-55.). We have revised corresponding text on Page 5, Lines 161-165.

Point 7: Again, for the histopathology evaluation of inflammation, were the samples blinded before visualization?  What grading scale was applied? Who performed the evaluation?

Response 7: Thanks for your comment. After H&E staining was performed, the severity of intestinal inflammation was evaluated by a pathologist in a blinded manner. Colonic histological scored was evaluated based on the severity of crypt depletion and distortion (0-3, from no crypt damage to entire crypt lost), degree of inflammatory infiltration (0-4, from no infiltration to large amount of infiltration) and the area of involvement (0:0, 1:1-25%, 2:26%-50%, 3:51%-75%, 4: 76%-100%). The histological score is the sum of each individual score. The histological evaluation of colon tissues was conducted by an independent investigator in a blinded manner. The grading scale used was supported by a previous study (Dieleman, L.A. et al. Chronic experimental colitis induced by dextran sulphate sodium (DSS) is characterized by Th1 and Th2 cytokines. Clin. Exp. Immunol. 1998, 114, 385-391.). We have added the corresponding description of histological evaluation on Page 5, Lines 178-180.

Point 8: For antibodies, clarify whether those were monoclonal or polyclonal.  Also, include details on secondary antibody and detection methods.

Response 8: Polyclonal antibodies were used. A goat anti-rabbit IgG-HRP was used as the secondary antibody. Enhanced chemiluminescence was used as the detection method. The related information was supplemented on Page 5-6, Lines 193-195.

Point 9: From the list of primers, it appears that you applied beta actin as your sole housekeeping gene, and used the delta-delta Ct method for quantitation.  This is a subpar approach to qPCR.  One, beta actin is typically expressed at much much higher levels than these other genes, making most PCR assays highly efficient. The delta-delta Ct method for quantitation does not account for the differing PCR efficiencies for different assays.  Rather, employ multiple HSKGs (at minimum 2, preferably with amplification at near the same Ct values as your target genes) for normalization and use the Pfaffl method for quantitation after generating PCR standards for each assay.

Response 9: Thanks for your comments. When we performed the qPCR, both β-actin and 18S rRNA had been amplified. But when calculating the relative expression of the target genes, we only used β-actin as the reference gene. The selection of 18S rRNA was based on a previous study that determined very similar target genes with ours (Hernández-Chirlaque, et al. Germ-free and antibiotic-treated mice are highly susceptible to epithelial injury in DSS colitis[J]. Journal of Crohns & Colitis, 2016, 10:1324.). Thus, according to the reviewer’s comments, both β-actin and 18S rRNA were used to re-calculate the relative expressions of the target genes, based on the Pfaffl method. Genes encoding β-actin and 18S were used as housekeeping genes (HSKGs). The HSKG with more similar PCR efficiency was used for the quantification of the target genes. All the data involved in the relative expressions of mRNA were reanalyzed and the figures were redrawn. The revisions were supplemented on Page 6, Line 204-211, Page 10, Figure 4 and Page 12, Figure 6

Point 10: For the microbiota sequencing, reference 24 does not provide thorough details.  Please include those details here or cite the appropriate references directly with all the necessary details for someone to replicate your work, including processing the data. 

Response 10: Thanks for your suggestion. For microbial DNA extraction and 16S rRNA sequencing, a reference which directly described the detailed method was cited. For bioinformatic analysis, the detailed method was provided. The revisions were supplemented on Page 6, Lines 214 and Page 6, Lines 219-228.

Point 11: Statistical analyses may not be appropriate if the animals were group housed.  Mice are coprophagic and largely share microbiomes within a cage. If they were group housed, your statistics should instead employ a mixed model with cage as a nested factor.  Moreover, you did not describe any data validation analyses, such as tests for normal distribution or equal variance, which are underlying assumptions with the ANOVA.  Were any transformations applied to the raw data?  Also, as I see results presented in figure 2, you clearly have data with two main factors:  time point and experimental diet.  As you are clearly looking at diet at different time points, then your statistical model must include both main factors (and their interaction).  You may need to consult with a statistician to reanalyze your data.

Response 11: We gratefully appreciate your valuable suggestions. The data which were not normally distributed or did not display equal variance were logarithmically transformed. This information was supplemented on Page 6, Line 235-237.

Two-way ANOVA which considers time and different treatments as two main factors was used to reanalyze the data in Figure 2B and Figure 2C. The revisions were supplemented on Page 6, Line 232-233, Page 7, Figure 2C, and Page 7, legend of Figure 2. Significant difference in body weight between AOM/DSS+millet and AOM/DSS+rice mice was shown only at the last time point. Thus, Figure 2B was not revised.

In the present study, mice were housed in groups of five mice per cage. According to your suggestion, we tried to rework the statistical analyses with a generalized linear mixed model which considers cage as a random effect and the treatment as a fixed effect. Unfortunately, due to the fact that some of the sample size within a single cage did not meet the criterion of n>3, the model is not suitable for the statistical analyses of the present study. Thus, ANOVA (for Physiological indexes) or Kruskal-Wallis with Dunn’s test (for bioinformatic analysis, according to your suggestion Point 22) that did not consider cage as a nested factor were still used in the present version of manuscript. Instead, according to a previous study (Sougiannis AT, et al. Impact of 5 fluorouracil chemotherapy on gut inflammation, functional parameters, and gut microbiota. Brain Behav Immun. 2019; 80:44.), it is ensured that at least one sample from each cage was used to partially prevent a cage effect. Also, a description discussing this limitation was supplemented in the manuscript. The revisions were supplemented on Page 6, Lines 228 and Page 16, Lines 535-538.

The current statistical analyses have been used in several of the previous studies which have a similar animal design with the present study (Kimura I, et al. Maternal gut microbiota in pregnancy influences offspring metabolic phenotype in mice[J]. Science, 2020, 367: eaaw8429; Sougiannis AT, et al. [J] Brain Behav Immun. 2019;80: 44.). We hope that the present statistical method could support the conclusion of the present study. We will try our best to improve the quality of the manuscript.

Point 12: Figure 2A This plot infers that some mice were lost during the study. What were the criteria for euthanasia?

Response 12: Thanks for your comments. During the experiment, the body weight and death situation of mice were observed and recorded every other day. Percent weight loss relative to the baseline is used as a surrogate measure of colitis severity. Mice with more than 20% body weight loss, hunched posture, and limited movement were euthanized to avoid excessive discomfort to animals as per institutional animal care committees. The criteria for euthanasia were supplemented on Page 4, Line 138-140.

Point 13: What statistical test was applied for survival analysis?  What is meant by survival "amount"?  Do you mean the number of animals?  If so, does that mean that by the end of the study, your sample size for the AOM/DSS+rice group was only 3?  

Response 13: Thanks for your comments. Survival analysis was conducted by Log-rank (Mantel-Cox) test. The manuscript has been revised to provide this information. Survival "amount" means the number of mice alive per group and was revised to "survival rate" to avoid misunderstanding. The revisions were supplemented on Page 6, Line 233-234, Page 7, Figure 2A and the legends.

By the end of the study, 3 mice were left in the AOM/DSS group. Except for these three mice, one mouse met the criteria for euthanasia two days before the end of the study. The mouse was sacrificed one day before the endpoint and was included into the samples. Thus, the sample size for AOM/DSS+rice group was 4. The manuscript was revised to supplement this information on Page 4, Line 141-143.

Also, the sample size of each group was supplemented in all the figure legends according to your suggestion (see details in Point 15).

Point 14: Please provide food intake data to aid in interpretation of the results.

Response 14: Thanks for your suggestion. During the experiment, the daily food consumption was calculated based on the total food consumption of the mice within the same cage. The manuscript was revised to provide this data. The revisions were supplemented on Figure S1 in the supporting information, and Page 7, Lines 248-249.

Point 15: As of figure 2, we still do not know how many mice per treatment group. 

Response 15: Thanks for your kind comments. There were 10 mice in each group at the beginning of the experiment. At the end of the study, the sample size of NM group, AOM/DSS group, AOM/DSS+millet group, and AOM/DSS+rice group were 10, 6, 7 and 4, respectively. We have added the number of mice in each group on the legends of all the figures.

Point 16: Tumor incidence is the number of mice with tumors (any number of tumors).  Please label Fig 2D correctly on the Y axis, as the values shown are multiplicity.  Also, did all mice exposed to AOM/DSS in this model develop tumors?  For panel D, the legend indicates that means with different letters are significantly different, however <3mm for AOM/DSS+millet does not appear different than >3mm AOM/DSS+rice.  I suspect separate analyses were applied for different tumor sizes.  Again, here you have two factors:  tumor size and diet treatment.  Please clarify these results and apply the proper statistical test.

Response 16: Thanks for your comments. "Tumor incidence" was revised into "tumor number" in figures, legends, and the text. All mice exposed to AOM/DSS treatment developed tumors, and the description was supplemented in the text. For Figure 2D, we intended to separately analyze the significance of tumor number with diameter <3mm and >3mm. According to your suggestion, we carefully studied the principles of statistical analysis, checked the data, and compared the statistical method with several of the previous studies (Khan I, et al. Mushroom polysaccharides and jiaogulan saponins exert cancer preventive effects by shaping the gut microbiota and microenvironment in Apc mice[J]. Pharmacological Research, 2019, 148:104448.). We thought that tumor sizes are different types of values measured rather than a factor. Thus, the statistical method for Figure 2D was not revised. However, the figure was redrawn to avoid misunderstanding of the statistical analysis. The revision was supplemented on Page 7, Figure 2D Line 254, and Line 263.

We hope that the current statistical method could properly present the results. If not, we will try our best to improve the quality of the manuscript.

Point 17: The authors should consider calculating tumor burden, the total tumor volume for each individual mouse. Also, the authors seem to overlook the lack of difference for the AOM/DSS+millet compared to the control AOM/DSS group.  The statement line 204 might be an over-exaggeration given that large tumors seem unaffected, which would be concerning as these are likely the more aggressive or advanced tumors.

Response 17: Thanks for your comments. The volume of tumor burden was calculated and added to the manuscript. The method of calculation was supplemented on Page 5, Lines 159-162. The results were supplemented on Page 7, Figure 2E and the figure legend, and the corresponding statement were revised on Page 7, Lines 257-259.

The over-exaggerated statement was removed and the current description was focused on the comparison between millet-rich diet and rice-rich diet. The revision was supplemented on Page 7, Lines 259-260.

Point 18: For H&E images, how many fields of view were examined for each colon?  Are these representative images?

Response 18: Thanks for your comments. At least 3 random fields of view were photographed at 100× magnification with a microscope. We chose these representative images from each group based on histological evaluation. The corresponding text was revised on Page 5, Lines 176-177, and Page 9. Line 308.

Point 19: For the protein and transcript analyses, how do you account for the presence of tumor tissue in those samples.  It seems to me that you have patterns of gene expression that likely reflect the tumor state, rather than earlier molecular events that lead to tumor development that were influenced by the diet. With some colon tissues having higher tumor burden than other, how do you know that these apparent differences in expression are related to the dietary treatments rather than differences in the abundance of cancer cells in the colon samples?  VEGF and PCNA are good examples; one would rightly expect their expression to be highly correlated with tumor abundance. This flaw is the main problem of animal studies designed this way, where samples are collected only at the end when disease is manifest, rather than collecting cohorts of samples throughout the study to understand dynamic processes leading to disease development.

Response 19: Thanks for your valuable comments. At the end of study, the mice were sacrificed and the whole colon tissues were removed. The distal colon tissues (5 mm) were used for histological analyses. The remaining colon tissues were cut in half lengthwise and was wholly homogenized before mRNA and protein analyses.

According to your suggestion, the corresponding statement "millet treatment could regulate the downstream signaling proteins associated with cell growth, proliferation, and survival" was removed from the manuscript. To collect the samples at multiple time points or to collect both tumor tissues and normal tissues in different groups is indeed a better way to understand the dynamic processes leading to CRC. Thus, a description of the limitation was supplemented in the Discussion section. The revisions were supplemented on Page 11, Line 362-364 and Pages 14-5, Lines 472-478.

A colitis-associated colorectal cancer model was used in the present study. The levels of cytokines, FOXP3, COX-2, iNOS, and the phosphorylation of STAT3 was thought to be earlier molecular events that influence the progression CRC. Moreover, the mRNA expressions of gut receptors GPCRs and AHR, and the microbiota metabolites of fiber and tryptophan could be directly influenced by different diets. These molecules were reported to influence the progression of IBD and CRC. The study design in the present study has also been used in several previous studies investigating the effect of dietary treatment on CAC ([1] Tian, Y. et al. Short-chain fatty acids administration is protective in colitis-associated colorectal cancer development. J Nutr Biochem 2018, 57, 103-109. [2] Wu, X. et al. Chemopreventive Effects of Whole Cranberry (Vaccinium macrocarpon) on Colitis-Associated Colon Tumorigenesis. Mol Nutr Food Res 2018, 62, e1800942.).

Again, great thanks for your valuable suggestion. We hope that the current version of manuscript could met the criteria of Nutrients. And we will try our best to improve the quality of the manuscript.

Point 20: Please plot rarefaction curves as means + error for easier visualization.  Also, please be consistent in coloring and graph style to match the prior plots.  That bright yellow is hard to see on a white background.  Also the legend is incomplete, in that the symbols are not explained.  Why no error bars for bar graphs for 7A and 7B?

Response 20: Thanks for your suggestion. The plot rarefaction curves has been revised to use a means + error manner. All the figures have been revised so that the graph style and color are consistent. The bright yellow color has been replaced by violet for easier visualization. The explanation of the symbols was added in the figures or the corresponding legends. The revisions were supplemented on Page 13, Figure 7, and Lines 420-421.

Point 21: Was alpha diversity for the millet diet group different than the rice group?

Response 21: There was not significant difference in alpha diversity between millet diet group and rice group. A description was supplemented on Page 12, Lines 410-412.

Point 22: Statistical analyses appear to have been performed for the bacteria taxonomic data.  However, an ANOVA is not the right test, as these data are very likely to be zero-inflated with a left shifted distribution.  Consider using a tool like Microbiome Analyst to help work with these data. https://www.microbiomeanalyst.ca/faces/home.xhtml

Response 22: Thanks for your comments. For the bioinformatic analysis, the statistical analysis was reworked. The Kruskal-Wallis with Dunn’s test was used to assess significant differences between the groups according to several previous studies (Ang, Q.Y. et al. Ketogenic Diets Alter the Gut Microbiome Resulting in Decreased Intestinal Th17 Cells. Cell 2020, 181, 1263.), and the description of statistical analysis used was added. Microbiome analysis was performed with I-Sanger, a real-time interactive online platform for data analysis (https://www.i-sanger.com/). The revisions were supplemented on Page 6, Lines 234-235, Page 13, Fig 7 and the corresponding legends.

Point 23: Comparing proportions between only the millet vs. rice doesn't make much sense.

Response 23: Thanks for your kind comments. Considering your suggestion, we have replaced the Figure 7D (Difference between AOM/DSS+millet group and AOM/DSS+rice group on microbial community) with a heatmap analysis among four group to provide more information. Also, the description of the figure was supplemented. The revisions were supplemented on Page 13, Figure 7D and Page 14, Lines 423-430.

Point 24: Discussion, line 358-360 reads as overly broad regarding the main endpoints. At the end of the study, DAI was not different between the millet and rice diets. And, while the millet diet did lead to fewer tumors (both small and large) compared to the rice diet, that was only the case for the small tumors compared to the standard diet. It's not known whether the overall tumor burden was different.  Line 360 - tumor incidence data were not shown.  Overall, this first paragraph is an overstatement of these findings.

Response 24: Thanks for your valuable comments. The supplement of millet reduced the levels of DAI, number of tumors (>3 mm or <3 mm) and tumor burden, compared to rice supplement, and reduced the level of tumor burden compared to the AOM/DSS group. In the previous version of manuscript, we intended to discuss the difference between millet supplement and rice supplement.

The manuscript has been revised to avoid misunderstanding and exaggeration. Also, some description was added to discuss the difference between millet-supplement group and the AOM/DSS group, based on the reduced level of tumor burden. The description of tumor incidence was removed. The revisions were supplemented on Page 14, Lines 436-443.

Point 25: The discussion fails to address any limitations of this study, of which I have identified quite a few that make interpretation of the results quite challenging.

Response 25: We greatly appreciate your valuable suggestions. According to your suggestion, the limitation of this study was discussed on Page 14-15, Lines 472-478 and Page 16, Lines 535-538.

Reviewer 2 Report

Abstract: Study design is not well explained. I thought it was only two groups, but the authors did a great job with study design, therefore that should be mentioned briefly in the abstract. please include about microbiome data, this is very interesting and it should capture the readers attention to this manuscript. Conclusion doesn't seem to be appropriate with the design/model as this is more advanced CRC model rather than early stage. Please make appropriate changes. 

Introduction: great flow and nicely done. Why did the authors choose this model? Please give a rationale to why this model with 3 DSS treatments at 2%.

Methods: figure one is not very clear. I would suggest remaking the figure into a flow diagram to better understand the treatments. It seems like on the text the diet started on week 2? But on the diagram is not very clear and it seems like the diet was for the entire period of the study, even when the AOM was administered. I would also suggest putting the n per group, I'm assuming it was n=10 per group. Was that powered to previous studies? Considering survival rate, it seems like the n will be much smaller when mice finish the study. 

Since this is a diet specific study, I would suggest putting the diet information in the manuscript rather than the supplementary. 

Where fecal samples collected at baseline?

Please add a section for tumor counts/measures and how it was done, since the results are being shown. 

Results:The figures resolution needs to be improved. It is not clear what is writing on the axis and the superscripts letter on the graphs. 

3.2 again please add the diet data to the manuscript.

3.3 please fix figure resolution as it is hard to see the details on the axis. Where any histopathology score done? Or maybe staining for collagen tissue which is commonly observed in DSS model. 

3.4 and 3.5, please fix the resolution of the figures. 

Page 10, if feels like that paragraph below the figure is out of context or missing other parts. Please fix. same for the paragraph on page 11. There is not sign of figure 7 being cited on the text. 

3.7 please mention one of these results in the abstract as they seem relevant to this model and study 

Discussion: authors need to discuss the reduced number of animals in the study. It seems that the n was reduced from the beginning and that is not clear in the results and discussion. If in fact the n is different and the analysis, please specify that in the figure legends. 

Where is figure 8 being discussed here? Please cite that on the text. 

It is not clear the association the authors make on the paragraph in page 13 about the control diet having the same dietary fiber content than the millet diet. Why is that? Shouldn't that have been part of the study design, and in fact evaluate the effects of a high fiber cereal, millet, in this model? Please elaborate that more, otherwise the control is not a control. 

Please discuss why NM group has high SCFA compared to other groups. If it is because of the high fiber, please elaborate, because to me this becomes a bit contradictory as a control group. Otherwise, the authors can easily compared the millet to the control, the control being a good (high fiber diet?). 

Great study and very interesting. Looking forward to seeing the edits and improvements of this manuscript. 

Author Response

Response to Reviewer 2 Comments

Dear Editors and Reviewers:

Thank you very much for your letter and comments on our manuscript entitled “Dietary supplementation of foxtail millet ameliorates colitis-associated colorectal cancer in mice by activation of gut receptors and suppression of STAT3 pathway” (ID: nutrients-869563). We greatly appreciate the constructive comments that are very helpful for our revision of the manuscript. We have made additions and corrections according to your comments. The revised sections in the manuscript are highlighted in red.

We do hope that the revised manuscript adequately addressed your comments. The major corrections and the responses to the reviewers’ comments are as follows:

Point 1: Abstract: Study design is not well explained. I thought it was only two groups, but the authors did a great job with study design, therefore that should be mentioned briefly in the abstract. please include about microbiome data, this is very interesting and it should capture the readers attention to this manuscript. Conclusion doesn't seem to be appropriate with the design/model as this is more advanced CRC model rather than early stage. Please make appropriate changes. 

Response 1: Thanks for your valuable suggestion. We have rewritten the Abstract according to your suggestion, the corresponding revisions were made on Page 1, Lines 22-24, 27-28 and Lines 31-34.

Point 2: Introduction: great flow and nicely done. Why did the authors choose this model? Please give a rationale to why this model with 3 DSS treatments at 2%.

Response 2: Thank you for your kind comments. The AOM/DSS mice model is widely used in mimicking the development of human colitis-associated colorectal cancer. AOM was used to induce carcinogenesis, and then the mice were continuously exposed to the inflammatory stimulus of DSS, thus, an ulcerative colitis model of carcinogenesis could be established. The combination of the mutagenic agent azoxymethane (AOM) and dextran sodium sulfate (DSS) can accelerate the formation of CAC. Moreover, three cycles of DSS administration could increase the incidence of colon tumorigenesis. ([1] Neufert, C.; Becker, C.; Neurath, M.F. An inducible mouse model of colon carcinogenesis for the analysis of sporadic and inflammation-driven tumor progression. Nat Protoc 2007, 2, 1998-2004. [2] Shan, S.; Wu, C.; Shi, J.; Zhang, X.; Niu, J.; Li, H.; Li, Z. Inhibitory Effects of Peroxidase from Foxtail Millet Bran on Colitis-Associated Colorectal Carcinogenesis by the Blockade of Glycerophospholipid Metabolism. J. Agric. Food Chem. 2020, 10.1021/acs.jafc.0c03257.).

Besides, different mouse strains show different susceptibility to DSS. It is generally considered to be safe to control the DSS concentrations between 1-3% (mass/volume), but are also dependent on the mouse strain. BALB/c was chosen as the experimental animals in this study, and several related researches were referred for the selection of a proper dose of DSS ([1] Sougiannis, A.T.; VanderVeen, B.N.; Enos, R.T.; Velazquez, K.T.; Bader, J.E.; Carson, M.; Chatzistamou, I.; Walla, M.; Pena, M.M.; Kubinak, J.L., et al. Impact of 5 fluorouracil chemotherapy on gut inflammation, functional parameters, and gut microbiota. Brain Behav Immun 2019, 80, 44-55. [2] Song, H.; Wang, W.; Shen, B.; Jia, H.; Sun, Y. Pretreatment with probiotic Bifico ameliorates colitis‐associated cancer in mice: Transcriptome and gut flora profiling. Cancer Sci. 109.).

The rationale for the establishment of the model was supplemented on Page 2, Lines 65-69. And a reference was cited to support the use of 2% DSS on Page 3, Line 116.

Point 3: Methods: figure one is not very clear. I would suggest remaking the figure into a flow diagram to better understand the treatments. It seems like on the text the diet started on week 2? But on the diagram is not very clear and it seems like the diet was for the entire period of the study, even when the AOM was administered. I would also suggest putting the n per group, I'm assuming it was n=10 per group. Was that powered to previous studies? Considering survival rate, it seems like the n will be much smaller when mice finish the study. 

Response 3: Thank you for your suggestion. We apologize for the inconsistency between the graph and the text. In fact, the diet was used for the entire period the study. We have revised Figure 1 on Page 3 according to your comments.

At the beginning of the study, the sample size of each group was 10, which was consistent with several previous studies ([1] Liu, L.Q. et al. Tea Polysaccharides Inhibit Colitis-Associated Colorectal Cancer via Interleukin-6/STAT3 Pathway. J. Agric. Food Chem. 2018, 66, 4384-4393. [2] Chung, K.S. et al. Chemopreventive Effect of Aster glehni on Inflammation-Induced Colorectal Carcinogenesis in Mice. Nutrients 2018, 10.). Since the chemical reagents for the establishment of the AOM/DSS-induced CRC model was toxic to mice, some of the animals were sacrificed during the study. The sample size was supplemented in the legends of all the figures (details discussed in Point 13).

Point 4: Since this is a diet specific study, I would suggest putting the diet information in the manuscript rather than the supplementary. 

Response 4: Thank you for your suggestion. We have added the diet information on Page 4, Lines 145.

Point 5: Where fecal samples collected at baseline?

Response 5: Thank you for your comment. The fecal samples were collected in the SPF laboratory environment. Fecal samples were collected immediately following defecation, transferred into sterile cryotubes, and snap-frozen in liquid nitrogen. The corresponding text were revised on Page 4, Lines 136-138.

The fecal sample at the beginning of the study was not collected. Instead, the change of gut microbiota induced by CRC was determined by the comparison between the NM group and the AOM/DSS group. The change of gut microbiota induced by the diet supplemented with millet and diet supplemented with rice was determined by the comparison between the diet-supplement groups with the AOM/DSS group. A heatmap analysis was supplemented to compare the difference in microbiota between groups. The corresponding revisions were supplemented on Page 13, Figure 7D. We hope that the current results could support the conclusion of the present study.

Point 6: Please add a section for tumor counts/measures and how it was done, since the results are being shown. 

Response 6: Thank you for your suggestion. The description of tumor evaluation was added on Page 5, Lines 154-165.

Point 7: Results: The figures resolution needs to be improved. It is not clear what is writing on the axis and the superscripts letter on the graphs. 

Response 7: Thank you for your suggestion. The figures were revised to improve the resolution and to increase the size of letters. The corresponding changes were made on Page 7, Line 261.

Point 8: 3.2 again please add the diet data to the manuscript.

Response 8: Thank you for your suggestion. We have added the information of the diet on Page 4, Lines 145 and Page 8, Line 284.

Point 9: 3.3 please fix figure resolution as it is hard to see the details on the axis. Where any histopathology score done? Or maybe staining for collagen tissue which is commonly observed in DSS model. 

Response 9: Thank you for your valuable suggestion. Colonic histological scores was evaluated based on the severity of crypt depletion and distortion (0-3, from no crypt damage to entire crypt lost), degree of inflammatory infiltration (0-4, from no infiltration to large amount of infiltration) and the area of involvement (0:0, 1:1-25%, 2:26%-50%, 3:51%-75%, 4: 76%-100%). The histological score is the sum of each individual score. The histological evaluation of colon tissues was conducted by an independent investigator. The evaluation of histopathology score was conducted according to the previous study (Dieleman, L.A. et al. Chronic experimental colitis induced by dextran sulphate sodium (DSS) is characterized by Th1 and Th2 cytokines. Clin. Exp. Immunol. 1998, 114, 385-391.). Special thanks to your kind comment and we have supplemented the corresponding content on Page 5, Line 178-180 and Page 9, Figure 3G(e). We hope that the current results could support the conclusion of the present study.

Point 10: 3.4 and 3.5, please fix the resolution of the figures. 

Response 10: Thanks for your suggestion. We have fixed the resolution of Figure 4 (Page 10) and Figure 5 (Page 11).

Point 11: Page 10, if feels like that paragraph below the figure is out of context or missing other parts. Please fix. same for the paragraph on page 11. There is not sign of figure 7 being cited on the text. 

Response 11: Thank you for your comment. The position of the figure has been adjusted. Also, we have cited Figure 7 in the corresponding text on Page 12, Line 406.

Point 12: 3.7 please mention one of these results in the abstract as they seem relevant to this model and study. 

Response 12: Thanks for your advice. The description of the microbiome results was added in the abstract on Page 1, Lines 31-33.

Point 13: Discussion: authors need to discuss the reduced number of animals in the study. It seems that the n was reduced from the beginning and that is not clear in the results and discussion. If in fact the n is different and the analysis, please specify that in the figure legends. 

Response 13: Thanks for your comments. There were 10 mice in each group at the beginning of the experiment. At the end of the study, the sample size of NM group, AOM/DSS group, AOM/DSS+millet group, and AOM/DSS+rice group were 10, 6, 7 and 4, respectively. During the study, mice with more than 20% body weight loss, hunched posture, and limited movement were euthanized to avoid excessive discomfort to animals as per institutional animal care committees. The criteria of the euthanasia were supplemented on Page 4, Line 138-140. A description discussing the decreased number of mice was supplemented on Page 14, Line 449-451. The number of mice per treatment group was supplemented on the legends of all the figures.

Point 14: Where is figure 8 being discussed here? Please cite that on the text. 

Response 14: Thanks for your suggestion. Figure 8 exhibited the underlying molecular mechanism of millet ameliorated colitis and inhibited the progression of CRC. We have cited this figure on Page 15 Line 501-502.

Point 15: It is not clear the association the authors make on the paragraph in page 13 about the control diet having the same dietary fiber content than the millet diet. Why is that? Shouldn't that have been part of the study design, and in fact evaluate the effects of a high fiber cereal, millet, in this model? Please elaborate that more, otherwise the control is not a control. 

Response: Thanks for your comments. When designing the in vivo study, we wished to compare the effects of millet-rich diet and rice-rich diet on colorectal cancer, and to further explore the molecular mechanism. Firstly, we analyzed the component of millet and rice. The content of dietary fiber and tryptophan showed significant difference between the two grains. It has been reported that the gut microbiota metabolites of dietary fiber and tryptophan may have protective effects against intestinal inflammation. During the study design, we tried to correlate the content of dietary fiber and tryptophan in grains and the beneficial effects (e.g., activation of gut receptors). After calculating the content of dietary fiber and tryptophan in corn starch (a major component in the original AIN 93G diet), millet and rice, the original addition of dietary fiber was reduced from 6% to 3% in the grain-supplemented modified AIN-93G diet. The final content of dietary fiber in the original AIN-93G diet (AOM/DSS group) and the millet-supplemented diet was both 6%, and was higher than that of the rice-supplemented diet. The content of tryptophan was higher in the millet-supplemented diet, and was relatively similar in the original AIN-93G diet and the rice-supplemented diet. Thus, by comparing the association between mRNA expressions of gut receptors and the content of grains, it can be further deduced that whether the content of dietary fiber and tryptophan is a key factor that influence the activation of gut receptors.

In addition, from the comparison between AOM/DSS mice and the millet-treated mice, the beneficial effects of major nutrients (except for dietary fiber) and bioactive compounds in millet could be revealed. However, as the reviewer mentioned, strictly speaking, the AOM/DSS group is not a control.

Thus, the manuscript has been revised to remove the statement of "control group". The reason of the designing of diet was added in the Method. The revisions were supplemented on Page 3-4, Line 125-131 and Page 16, Line 504-511.

Point 16: Please discuss why NM group has high SCFA compared to other groups. If it is because of the high fiber, please elaborate, because to me this becomes a bit contradictory as a control group. Otherwise, the authors can easily compare the millet to the control, the control being a good (high fiber diet?). 

Response: Thanks for your comments. In the present study, the levels of SCFAs in the AOM/DSS group, AOM/DSS+millet group, and AOM/DSS+rice group were lower than those of the NM group. The results are consistent with several of the previous studies (Peng Y, et al. Gut microbiota modulation and anti-inflammatory properties of anthocyanins from the fruits of Lycium ruthenicum Murray in dextran sodium sulfate-induced colitis in mice[J]. Free Radical Biology and Medicine, 2019, 136:96; Hu, Q, et al. Dietary Intake of Pleurotus eryngii Ameliorated Dextran Sulfate sodium-induced Colitis in Mice. Molecular Nutrition & Food Research 2019, 63:1801265). The reason was supposed to be that the change of gut microbiota produced by AOM/DSS treatment is a more important factor than the content of fiber intake that influence the fermentation of fiber. Thus, we mainly compared the millet-treated mice with the AOM/DSS mice to eliminate the effect of AOM/DSS on the SCFAs-producing effect. The revisions were supplemented on both the Results and Discussion sections on Page 11, Line 378-379, Page 14 Line 423-427 and Page 16, Lines 523-526, to explain the reason why NM group had higher levels of SCFAs compared to other groups.

Round 2

Reviewer 1 Report

The revised paper is improved, though some significant issues remain, particularly regarding data visualization and presentation of the microbiome results.

One, what post-hoc tests were used to compare across diet groups in the ANOVA? 

Fig 2B,C - are the significance results reported for the last time point only? Or for main effects of the different diets for the duration of the study?

The brackets in Fig 2C are still unclear given the explanatory text in the legend.  Are they meant to indicate that for the time period indicated by the bracket those diet groups are different?  Typically brackets are used to point to specific data points, so it seems like you're pointing to wk 5 data compared to wk 9 data for the bracket with the # symbol.  It's important to use conventional notations, or if not, to be very clear in the legend.

Last, I'm still quite concerned about the statistical analyses of the microbiome study.  Use of a nonparametric test for examining differential abundance does not account for the fact that these data are likely very strongly influenced by group housing.  Now, given the PCA results, I do suspect that the diets influenced microbiome composition to a fairly remarkable extent.  However there is grouping in that data that causes concern.  For example, in the millet group, three data points are clustered and located very near samples from both the rice group and the AOM/DSS control group.  So, are mice from a single cage responsible for the apparent separation of those data?

While I recognize that other studies have been previously published that have not considered cage effects, including papers in some prestigious journals, that does not mean these issues are not a problem that can taint the scientific literature.  As a community of researchers working in the area of the gut microbiome, we need to strive to improve our study designs and presentation of the results.  

Given that it is likely not feasible for this team to go back and redo this study with a greater sample size to account for animal deaths and single animal housing to avoid confounding cage effects, then the authors need to reconsider the way in which they are presenting the data.

First, rather than plotting data as bar charts with errors, please plot each individual data point with the mean represented as a line and SD.  This would be preferred for most all of the data graphs in the paper, and would be a more robust approach for data visualization.

For the microbiome data, given the concerns about data analyses that cannot be addressed statistically (you're right, with only 2 cages per group, a mixed model will not work to account for random effects of cages), the authors must present the data for individual animals with identifiers that clarify which mice were group-housed within a diet group.  For example, they can expand the heat map shown in panel 7D to represent each individual mouse with bars across the top to represent groping by cage.  An unsupervised hierarchical clustering analysis should also be considered to determine whether animals group by diet treatment or cage.  (There is an online tool called ClustVis that integrates with R that will make this fairly easy to perform)

Reviewer 2 Report

Thank you for addressing the suggestions and comments. The manuscript has been highly qualified after the revisions. 

Author Response

Thanks for your time and valueable suggestions!Our article has been improved a lot with your help.